# Fermentation Regulation and Ethanol Production of Total Mixed Ration Containing Apple Pomace

**Jiachen Fang [1], Zhumei Du [2] and Yimin Cai [3],***

[1] Faculty of Agriculture and Life Science, Hirosaki University, Hirosaki 036-8561, Japan; fang@hirosaki-u.ac.jp
[2] College of Animal Science and Technology, Yangzhou University, Yangzhou 225009, China; duzhumei2021@126.com
[3] Japan International Research Center for Agricultural Sciences (JIRCAS), Tsukuba 305-8686, Japan
*    Correspondence: cai@affrc.go.jp

**Abstract:** To effectively utilize local fruit residue resources and regulate ethanol production in fermented feed, the impact of moisture adjustment, lactic acid bacteria (LAB) inoculant, and chemical additive on the fermentation characteristics and ethanol production of total mixed ration (TMR) containing apple pomace was studied. The TMR was prepared with apple pomace, corn, wheat bran, soybean meal, timothy, and alfalfa hay. The mixing proportion of apple pomace was 15% based on dry matter (DM). In experiment 1, the moisture in TMR was unadjusted (control) or adjusted to 45, 50, and 55%, respectively. TMR containing 55% moisture was used in experiment 2, and the treatments were control, homo-fermentative LAB (*Lactobacillus plantarum*, LP), hetero-fermentative LAB (*Lactobacillus buchneri*, LB), and calcium propionate (CaP). The laboratory-scale fermentation system was used to prepare TMR, and their fermentation characteristics were analyzed after 60 days of ensiling. In experiment 1, the pH of the various TMRs was around 4.1. As the moisture decreased, lactic acid increased ($p < 0.05$) and ammonia-N decreased ($p < 0.05$). The ethanol decreased significantly with moisture adjustment compared to the control and the TMR with 50% moisture had the lowest ethanol content ($p < 0.05$). In experiment 2, LP treatment increased lactic acid, and decreased acetic acid and ammonia-N significantly ($p < 0.05$), while LB treatment had no effect on fermentation. LP and LB each had no effect on the ethanol content. TMR treated with CaP significantly decreased the ethanol and acetic acid content ($p < 0.05$), but did not inhibit lactic acid production compared to control. The results confirmed that adjusting the moisture of TMR to 50% and adding CaP could effectively inhibit the excessive production of ethanol in TMR of apple pomace. Homofermentative LAB can better improve the fermentation quality of TMR than heterofermentative LAB, but neither can inhibit the production of ethanol. This is of great significance to the effective utilization of apple residue resources and the promotion of livestock production.

**Keywords:** apple pomace; ethanol production; microbial and chemical additive; moisture adjustment; total mixed ration

## 1. Introduction

As the world economy develops and the population continues to grow, the global demand for food production continues to rise. At present, many countries in the world, including Japan, are facing the problem of food security. With the global shortage of grain feed supply and rising feed prices, the effective use of food by-product resources, such as pomace, as livestock feed is considered to be one of the ideal solutions to this problem. Apple, citrus, banana, and grape are known as the four major fruits of the world. Apple pomace is a by-product that remains after apples are ground and pressed in the production of juice; in many countries, it is usually discharged in large quantities after the apples are harvested. In recent years, the total output of apple pomace in the world is estimated to be close to 4 million tons, and it will continue to increase in the future [1].

Relevant studies have shown that apple pomace has multiple uses, such as the production of aromatic substances, dietary fiber, citric acid, pectin, seasoning, and feed utilization [2,3]. However, the moisture in apple pomace is high and rich in sugar. If the discharged residue is not treated in time, it can easily lead to public health and environmental pollution such as corruption, pests, effluent, and unpleasant odor [1,2]. Dried apple pomace is easy to transport and store, but the drying process consumes a lot of energy, potentially from fossil fuels, resulting in high feed costs and impracticality. Therefore, fermented total mixed ration (TMR) is considered as an important feed preparation technique that can effectively utilize high-moisture food by-products such as apple pomace.

Generally, fermented TMR has good fermentation qualities, but silage or TMR containing apple pomace produces not only organic acids but also ethanol. Ethanol has a preservative effect when mixed into the feed at a level of 1–2% dry matter (DM) [4]. However, a high content of ethanol will lead to a loss of DM and energy in the feed [5], a reduction in feed digestibility in ruminants [6–8], and also affect the composition and flavor of milk [9], the birth weight of calves, deformity, and stillbirth [10].

The main producer of ethanol during ensiling is yeast, which is also one of the microorganisms that cause the aerobic spoilage of silage [11]. Moisture regulation will affect the microbial community dynamics and fermentation quality during ensiling [12,13]. Microbial inoculants, such as lactic acid bacteria, and chemical additives, such as propionic acid or its salts, can inhibit the proliferation of yeast and improve the aerobic stability of silage [14,15]. However, there is limited information on the effects of these preparation methods on the fermentation of TMR containing apple pomace.

In order to effectively utilize apple pomace resources to prepare high-quality TMR, the impacts of moisture adjustment, microbial inoculant, and chemical additive on the fermentation characteristics and ethanol production of total mixed ration containing apple pomace were studied.

## 2. Materials and Methods

### 2.1. TMR Preparation

Apple pomace is discharged in large quantities during apple juice production worldwide (Figure 1). The apple pomace used in this experiment consisted mainly of Fuji, Orin, Tsugaru, Jonagold, and Jonathan apples, etc., which were obtained from a fruit juice factory (Morita Apple Juice Company, Hirosaki, Japan) in December 2020. Apple pomace was used to prepare a TMR along with corn, wheat bran, soybean meal, timothy hay, alfalfa hay, and a vitamin/mineral supplement (Snow Brand Seed, Sapporo, Japan). When preparing the TMR, timothy and alfalfa hay were cut into 2 cm pieces and mixed with the other materials. The crude protein (CP) content and roughage proportion of the TMR were determined according to previous studies [16]. Table 1 shows the chemical composition of the materials and Table 2 shows the DM proportion of ingredients in the TMR and moisture adjustment detail for experiment 2.

**Table 1.** Chemical composition of total mixed ration materials used in this experiment.

| Item | DM (%) | OM | CP | EE | ADF | NDF | GE (kcal/kg DM) |
|---|---|---|---|---|---|---|---|
| | | (% DM) | | | | | |
| Apple pomace | 21.8 | 97.4 | 4.6 | 3.4 | 25.7 | 34.7 | 4.7 |
| Soybean meal | 86.2 | 92.7 | 51.8 | 1.4 | 8.5 | 14.5 | 5.0 |
| Corn | 85.4 | 98.7 | 9.0 | 2.6 | 3.6 | 16.1 | 4.7 |
| Wheat bran | 85.9 | 94.2 | 18.4 | 4.5 | 15.7 | 51.3 | 4.8 |
| Timothy hay | 88.3 | 93.5 | 7.1 | 0.4 | 41.3 | 65.6 | 4.6 |
| Alfalfa hay | 87.9 | 90.8 | 19.0 | 1.3 | 34.3 | 46.6 | 4.6 |

DM, dry matter; OM, organic matter; CP, crude protein; EE, ether extract; ADF, acid detergent fiber; NDF, neutral detergent fiber; GE, gross energy.

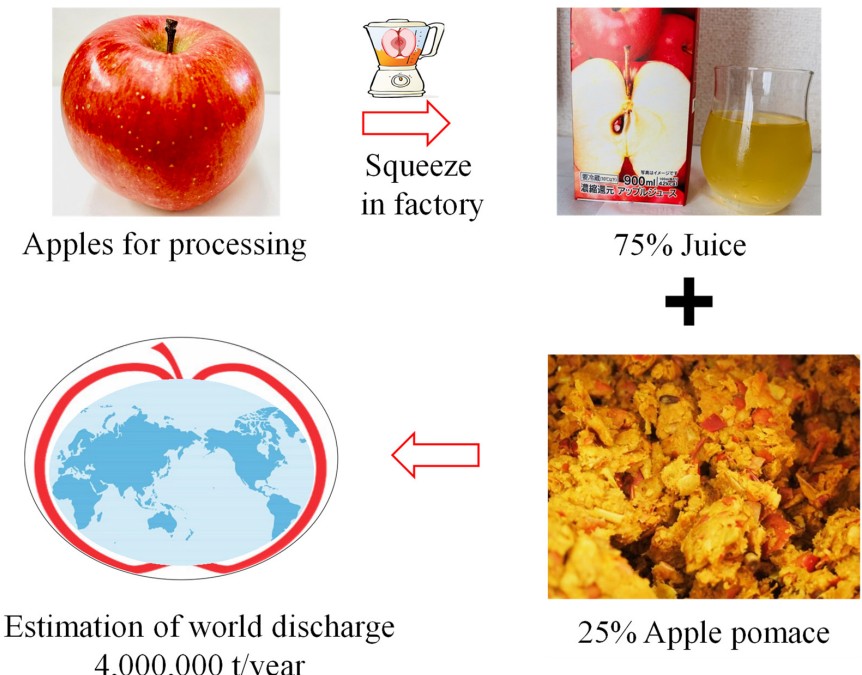

**Figure 1.** Production proportion and estimated world discharge of apple pomace.

**Table 2.** Ingredient proportions of TMR by dry or fresh matter.

| Material | Dry Matter (%) | Fresh Matter (%) | | | |
|---|---|---|---|---|---|
| | | **Control** | **M45** | **M50** | **M55** |
| Apple pomace | 15.0 | 41.6 | 38.1 | 34.6 | 31.1 |
| Soybean meal | 8.0 | 5.6 | 5.1 | 4.7 | 4.2 |
| Corn | 13.3 | 9.4 | 8.6 | 7.8 | 7.0 |
| Wheat bran | 12.9 | 9.0 | 8.2 | 7.5 | 6.7 |
| Timothy hay | 24.5 | 16.6 | 15.3 | 13.9 | 12.6 |
| Alfalfa hay | 24.9 | 16.9 | 15.5 | 14.1 | 12.7 |
| Vitamin/mineral supplement | 1.4 | 0.9 | 0.8 | 0.7 | 0.6 |
| Water | 0.0 | 0.0 | 8.4 | 16.7 | 25.1 |

TMR, total mixed ration; M45, M50, M55 represent the moisture content of TMR adjusted to 45, 50, and 55% of dry matter, respectively.

In experiment 1, the mix ration of apple pomace in TMR was 15% of DM. The moisture of TMR was designed to be 40% (control), 45% (M45), 50% (M50), and 55% (M55). There was no moisture adjustment in the control, while the other treatments were adjusted for moisture with distilled water, respectively. In experiment 2, the same TMR menu was formulated as in experiment 1 and the moisture was adjusted to 55%. The treatments were the control, homofermentative lactic acid bacteria (LAB) inoculant (LP, *Lactobacillus plantarum* Chikuso-1; Snow Brand Seed, Sapporo, Japan), heterofermentative LAB (LB, *Lactobacillus buchneri* 11A44; Pioneer EcoScience Co., Tokyo, Japan), and calcium propionate (CaP). LP, LB, and CaP were added at 5 mg/kg, 5 mg/kg, and 0.4 g/kg, respectively. Three kilograms of TMR were prepared for each treatment, packed equally into three plastic bag silos (ST1015; 300 × 450 mm, ASONE, Osaka, Japan), degassed, and sealed with a vacuum packaging machine (AliceV952S, ASONE, Osaka, Japan) for storage at 20–25 °C for 60 days.

*2.2. Sampling and Chemical Analysis*

After 60 days of ensiling, the three TMR bags for each treatment were unsealed and sampled for chemical analysis. For determining the moisture of TMR, 50 g samples were distilled using toluene for three hours and moisture was calculated according to previous

studies [17]. TMR samples for chemical composition analysis were dried in a fan-forced oven at 60 °C for 48 h and ground to pass through a 1 mm screen with a sample mill. The CP and ether extract (EE) contents were analyzed according to previous studies [18]. The organic matter (OM) was calculated as weight loss upon ashing. Neutral detergent fiber (NDF) and acid detergent fiber (ADF) were analyzed according to previous studies [19]. Heat stable amylase and sodium sulphite were used in the NDF procedure, and results are expressed without residual ash.

The fermentation quality of TMR was analyzed by using cold-water extract. After the ensiling silos were unsealed, 100 g of wet TMR sample was homogenized with 300 mL of distilled water and kept in a refrigerator at 4 °C for 24 h, as described by previous studies [20]. The filtrate pH was measured using a glass electrode pH meter (D-21; Horiba, Tokyo, Japan). The ammonia-N content of TMR was determined by steam distillation of the filtrates as described by previous studies [20] using the Kjeltech auto distillation equipment (2200, Foss Tecator, Hoganas, Sweden). The contents of lactic acid, acetic acid, propionic acid, and butyric acid were analyzed by HPLC method using Shodex RS Pak column (KC-811, Showa Denko K.K., Kawasaki, Japan), DAD detector (SPD-20A, 210 nm, Shimadzu Co., Ltd., Kyoto, Japan), eluent (3 mmol/L $HClO_4$, 1.0 mL/min), and temperature (40 °C). The ethanol content of the TMR was determined using the gasification balance method according to previous studies [21]: the sample of TMR liquid extract was put in a vial and warmed to 65 °C for 15 min, then 0.5 mL of the upper steam in the vial was sampled and injected to a gas chromatograph (G-5000A; Hitachi, Tokyo, Japan) equipped with a thermal conductivity detector and a G-5000 stainless column (3 mm × 2 m, Unisole F-200; GL Science, Tokyo, Japan). The analytical conditions were as follows: column oven temperature, 40 °C; injector temperature, 150 °C; detector temperature, 150 °C.

*2.3. Statistical Analysis*

Differences were considered significant at a threshold of $p < 0.05$. Statistical analyses were performed using SPSS ver. 25 (SAS Institute, Cary, NC, USA). Data on the fermentation quality and chemical composition of each TMR were analyzed using one-way analysis of variance. Statistical comparisons were made using Tukey–Kramer tests.

**3. Results**

The chemical composition of the TMR materials used in this experiment is shown in Table 1. Soybean meal, corn, wheat bran, timothy hay, and alfalfa hay had a DM content higher than 85%, and their OM content ranged between 90–99% of DM. However, the DM of apple pomace was less than 22% and its OM content was higher than 97% of DM. The EE content of apple pomace and wheat bran were 3.4–4.5% of DM, while the EE content of other TMR materials was less than 2.6%. The NDF and ADF contents of apple pomace were below 26% and 35% of DM, respectively, while they ranged between 14–17% and 3–9% of DM in soybean meal and corn, and 46–66% and 15–42% of DM in wheat bran and two hays, respectively. The TMR materials' gross energy (GE) ranged from 4.6 to 5.0 kcal/kg of DM.

The ingredient proportions of TMR based on dry or fresh matter are shown in Table 2. The TMR treatments of the control, M45, M50, and M55 were prepared with fresh raw materials, including apple pomace, soybean meal, corn, wheat bran, timothy hay, alfalfa hay, vitamin/mineral supplement, and water in different proportions, and the final proportions were adjusted for each treatment to keep the dry matter content of each material as consistent as possible.

Fermentation characteristics and chemical composition of moisture-adjusted TMR in experiment 1 are shown in Table 3. The moisture of the control, M45, M50, and M55 TMR samples was approximately 40, 45, 50 and 55%, respectively. The pH of the four fermented TMR samples were similar, all around 4.1. The content of lactic acid was the highest in the M45 treatment ($p < 0.05$), followed by the M50 treatment, and the M55 treatment was the lowest ($p < 0.05$) level, similar to the control. Acetic acid and ammonia-N contents increased with increasing TMR moisture and reached a peak in the M55 treatment ($p < 0.05$).

Propionic acid and butyric acid contents were below detection levels (<0.001 g/kg of DM) in all TMR samples. The ethanol production of TMR decreased ($p < 0.05$) in the order of control, M55, M45 and M50. All TMR samples were similar in chemical composition with approximately 94% OM, 15% CP, 2% EE, 25–26% ADF, and 41–43% NDF.

**Table 3.** Fermentation characteristics and chemical composition of moisture-adjusted TMR in experiment 1.

| Item | Control | M45 | M50 | M55 | SEM | *p*-Value |
|---|---|---|---|---|---|---|
| Fermentation characteristics | | | | | | |
| Moisture, % | 40.4 [d] | 45.2 [c] | 50.4 [b] | 55.1 [a] | 3.182 | <0.001 |
| pH | 4.1 | 4.1 | 4.1 | 4.1 | 0.006 | 0.073 |
| Lactic acid, % DM | 3.1 [c] | 3.6 [a] | 3.3 [b] | 3.2 [c] | 0.124 | 0.016 |
| Acetic acid, % DM | 0.8 [d] | 1.1 [c] | 1.4 [b] | 3.1 [a] | 0.512 | <0.001 |
| Propionic acid, % DM | ND | ND | ND | ND | - | - |
| Butyric acid, % DM | ND | ND | ND | ND | - | - |
| Ammonia nitrogen, % TN | 1.9 [c] | 2.2 [b] | 2.4 [b] | 3.1 [a] | 0.263 | 0.021 |
| Chemical composition, % DM | | | | | | |
| Organic matter | 94.0 | 94.1 | 93.8 | 93.9 | 0.060 | 0.149 |
| Crude protein | 14.8 | 15.0 | 14.9 | 15.0 | 0.040 | 0.343 |
| Ether extract | 2.2 | 2.0 | 2.2 | 2.1 | 0.060 | 0.432 |
| Acid detergent fiber | 25.7 | 24.9 | 26.4 | 25.2 | 0.320 | 0.824 |
| Neutral detergent fiber | 43.2 | 41.5 | 43.0 | 43.0 | 0.390 | 0.535 |
| Ethanol | 3.6 [a] | 0.9 [c] | 0.4 [d] | 1.9 [b] | 0.701 | <0.001 |

[a–d] Different letters within rows and within mix show significant differences ($p < 0.05$). TMR, total mixed ration; M45, M50, M55 represent the moistures of TMR, adjusted to 45, 50, and 55% of dry matter, respectively; SEM, standard error of the mean; DM, dry matter; ND, not detected; TN, total nitrogen.

The fermentation characteristics of TMR treated with microbial inoculant and chemical additive in experiment 2 are shown in Table 4. Compared to the control, the fermentation quality of LB-treated TMR did not change greatly, while LP-treated TMR had significantly ($p < 0.05$) increased lactic acid content, and decreased pH, acetic acid, and ammonia-N contents. CaP-treated TMR had no difference in lactic acid and ammonia-N contents, but significantly ($p < 0.05$) lower acetic acid content compared with the control. The contents of propionic acid and butyric acid were the same as in experiment 1 and were below detection levels (<0.001 g/kg) in all TMR. The ethanol production of LP and LB-treated TMR was similar to the control, and there was no great difference between the two kinds of LAB inoculants. However, CaP-treated TMR significantly ($p < 0.05$) inhibited ethanol production compared with control or other treatments.

**Table 4.** Fermentation characteristics of TMR treated with microbial inoculant and chemical additive in experiment 2.

| Item | Control | LP | LB | CaP | SEM | *p*-Value |
|---|---|---|---|---|---|---|
| Fermentation characteristics | | | | | | |
| Moisture, % | 55.4 | 55.2 | 55.4 | 55.1 | 0.075 | 0.141 |
| pH | 4.1 [a] | 3.9 [b] | 4.2 [a] | 4.1 [a] | 0.053 | 0.031 |
| Lactic acid, % DM | 3.3 [b] | 5.1 [a] | 3.3 [b] | 3.1 [b] | 0.459 | <0.001 |
| Acetic acid, % DM | 3.0 [a] | 0.7 [c] | 3.0 [a] | 1.6 [b] | 0.576 | <0.001 |
| Propionic acid, % DM | ND | ND | ND | ND | - | - |
| Butyric acid, % DM | ND | ND | ND | ND | - | - |
| Ammonia nitrogen, % TN | 2.9 [a] | 1.2 [b] | 2.9 [a] | 3.1 [a] | 0.436 | <0.001 |
| Chemical composition, % DM | | | | | | |
| Organic matter | 93.8 | 94.1 | 93.8 | 94.0 | 0.080 | 0.286 |
| Crude protein | 15.1 | 15.0 | 14.9 | 15.1 | 0.050 | 0.486 |
| Ether extract | 2.1 | 2.0 | 2.0 | 2.0 | 0.030 | 0.446 |
| Acid detergent fiber | 25.9 | 26.1 | 25.4 | 24.9 | 0.270 | 0.738 |
| Neutral detergent fiber | 42.2 | 41.6 | 42.3 | 41.9 | 0.160 | 0.637 |
| Ethanol | 1.8 [a] | 1.8 [a] | 1.7 [a] | 0.2 [b] | 0.431 | <0.001 |

[a–c] Different letters within rows and within mixes show significant differences ($p < 0.05$). TMR, total mixed ration; LP, lactic bacteria (*Lactobacillus plantarum*); LB, lactic bacteria (*Lactobacillus buchneri*); CaP, calcium propionate; SEM, standard error of the mean; DM, dry matter; ND, not detected; TN, total nitrogen.

## 4. Discussion

TMR is a completely mixed feed that mixes all essential nutrients, such as minerals and vitamins, in addition to roughage, such as grass, and concentrated feed, such as corn, to meet the nutritional needs of livestock such as dairy cows. Fermented TMR can efficiently utilize agricultural by-products and food residues to produce high-quality fermented feed at a low cost. Therefore, the development and utilization of apple residue TMR preparation technology can not only effectively utilize local feed resources and improve the production capacity for self-sufficient feed, but also play an important role in the sustainable production of livestock.

In this study, apple pomace, crop by-products such as soybean meal and wheat bran, and two kinds of hay were used to prepare TMR. Fresh apple pomace, usually discharged from juice plants, has a high moisture content, approaching that of high moisture forage crops or grasses. As shown in Table 1, the CP content of apple pomace was lower than that of timothy hay, while its EE content is higher than that of legume alfalfa. In addition, apple pomace is rich in sugars such as glucose and fructose, which can effectively promote silage fermentation. Therefore, apple pomace can be used as a raw material for TMR, which is a potential ruminant feed resource.

As shown in Table 2, alfalfa and timothy hay were mainly used as roughage sources for TMR. According to the chemical composition of various TMR raw materials and the nutritional requirements of ruminants, the proportion of roughage is set at about 50% of the TMR based on DM. Apple pomace has ADF and NDF contents higher than 25% and 34% of DM, respectively, and can be utilized as roughage for ruminant livestock. According to our previous study, ruminants should not be fed apple pomace with more than 20% of DM [6]. In this study, the mix proportion of apple pomace in TMR was 15% of DM, which was second only to the proportion of the two types of hay. In addition, the fat content of apple pomace was 3.4% of DM, which is usually higher than the hays used in this experiment. This may be due to the fact that water-soluble carbohydrates, e.g., sugars, are removed during the production of apple pomace, resulting in the high relative fat content of apple pomace [1].

In order to prevent the outflow of fermentation juice and improve transportation efficiency, feed factories usually adjust the moisture content to about 55% when preparing fermented TMR [12]. Previous studies have shown that changing the moisture content can affect the fermentation quality of TMR, but the results vary with the blending ratio of various materials and there is no causal relationship between TMR fermentation patterns and moisture [13]. To date, there is limited research on the effect of moisture adjustment of TMR feedstock on fermentation quality. In experiment 1, the fermentation quality of various TMRs (control, M45, M50, and M55) was good, the pH was 4.1, and the lactic acid content was higher than 3.1. According to Tudisco et al. [22], for all the these the fermentation was adequate, as suggested by the ammonia-N values lower than 7 g/kg total nitrogen. This is because the various raw materials of TMR are rich in nutrients, and their water content is between 40–55%, which is within the suitable moisture range for TMR fermentation. In the present study, all TMRs can be prepared with good quality. In experiment 1, the chemical components of OM, CP, EE, NDF, and ADF in each treatment of TMR did not change significantly, which also confirmed from the side that these TMR fermented well and effectively preserved various chemical components.

The content of acetic acid and ammonia-N in TMR tended to increase with the increase of water, which indicated that higher water content could increase the proliferation of aerobic microorganisms, and the aerobic environment in the early stage of TMR fermentation easily led to the production of acetic acid and the decomposition of protein. However, this period is generally short and will be quickly replaced by the anaerobic and acidic environment formed by lactic acid fermentation. Therefore, it usually does not have a profound adverse effect on the fermentation quality of TMR. However, moisture adjustment of TMR strongly affects ethanol production. The control with 40% moisture produced the most ethanol. The M50 treatment has the largest inhibitory effect, followed by the M45 treat-

ment, and the M55 treatment had the smallest inhibitory effect. Because this study did not measure the dynamics of microbial fermentation in the TMR, and there is a lack of research on yeast water activity, the reasons for this are unclear. Related studies have reported that yeast in silage produces ethanol during fermentation, and the growth of yeast is affected by various factors, including microbial diversity, community structure, metabolites, water activity, microbial permeability, and acidic cations in the silage environment [23]. From the experimental results of this study, it can be inferred that some epiphytic yeasts in apple pomace TMR may be affected by water activity and prefer to grow in lower or higher moisture ranges. Yeast, LAB, and other microorganisms coexist in fermented feed and form a symbiotic microbial network system. The interaction between microorganisms and their final metabolites will affect the fermentation quality of TMR and the physiological metabolism of livestock. As described in the introduction, 1–2% of DM ethanol content in fermented feed can inhibit spoilage and improve the palatability of livestock. However, alcoholic fermentation in TMR will cause DM and energy loss, and the high ethanol content in the feed will negatively impact milk quality and breeding cattle. In the future, it is necessary to isolate and taxonomically study the yeast of apple pomace TMR, and to deeply explore the water activity of the yeast and the mechanism of ethanol production.

Previous studies show that LAB activity decreased in TMR containing green tea residue. In contrast, acetate and ammonia-N producing bacteria activity increased with increasing moisture, resulting in decreased lactic acid and increased acetic acid and ammonia-N with increasing moisture [24]. In this study, the raw material composition and water content of TMR were different from those mentioned above, but the fermentation quality of apple residue TMR was basically consistent with the results of these studies. Acetic acid inhibited yeast growth more effectively than lactic acid, but ethanol production in the TMR was not affected by the acetic acid content in this study. This may indicate that the content of acetic acid produced during TMR fermentation did not reach the concentration that can inhibit the growth of yeast. There is limited information on the tolerance of acetic acid concentration, propionic acid and other organic acids in yeast associated with TMR fermentation, which can be used as a topic for future research.

A study reported the quality of TMR fermentation with different proportions of apple pomace and found that LAB may be inhibited when the ethanol concentration exceeds 2% of DM [6]. In experiment 1, the ethanol concentration in control was 3.6%. The higher ethanol in TMR of control may have inhibited LAB activity, resulting in lower lactic acid than in the M45 and M50 treatments. The lowest acetic acid level in TMR of control may be also caused by ethanol inhibiting the activity of acetic acid-producing bacteria. Regarding the ammonia-N, ethanol in TMR may also affect the activity of proteolytic bacteria, and previous studies reported that ammonia N in tofu-cake silage decreased with ethanol addition [16]. Thus, the observation of the lowest ammonia-N in the control TMR may have been due to the inhibition of protein degradation by ethanol during fermentation.

Based on the results of experiment 1, when preparing TMR, the proportion of apple pomace should be intermediate or low to guarantee room for moisture adjustment, while moisture in the range of 45–50% had a stronger inhibitory effect on ethanol. Although moisture level also affected the levels of lactic acid, acetic acid, and ammonia-N, their production was all in the good range, so the inhibitory effect on ethanol should be prioritized when adjusting the moisture content of TMR.

In experiment 2, LP increased lactic acid, decreased acetic acid and ammonia-N, and had no effect on ethanol, whereas LB has no effect on fermentation. Previous studies have shown that the compatibility of the silage material and inoculated strain determines the success of the microbial additive, and incompatible strains may have no effect on fermentation [25,26]. The LP and LB used in the experiment were manufactured for grass silage, and the fermentation result indicated that LP was compatible with apple pomace-based materials, whereas LB was incompatible. Previous studies showed that when LAB are used to inhibit ethanol in sugarcane, LB is more effective than LP because LP fermentation produces lactic acid only while LB fermentation produces lactic and acetic acid [27]. Since

lactic acid has a much weaker inhibitory effect on yeast than acetic acid, LB better inhibits ethanol than LP. This may explain that while LP significantly increased lactic acid, ethanol was not inhibited in this experiment. Previous studies reported that when wild strains isolated from feed crop were inoculated in the same material as the isolated strains, the results were often better than for commercial strains [25]. To develop LAB that can inhibit ethanol in apple pomace, the LAB strains in apple pomace should be analyzed to find effective strains. Yeast and LAB in TMR both use sugar as fermentation substrate and there is competition between them. Some researchers studied the fermentation quality of TMR containing apple pomace at different proportions. They found that lactic acid production peaked when the ethanol concentration was less than 2% DM but decreased significantly at ethanol concentrations above 2% DM [6]. The reason discussed was the inhibitory effect of ethanol on LAB. In experiment 2, the ethanol content of all treatments was less than 2% DM, so the lactic acid production may be mainly affected by the additive instead of the ethanol.

Propionic acid and its salts are chemical additives for food or feed that have antifungal effects. They are commonly used to inhibit the growth of yeasts and molds in silage, with the main purpose of increasing the aerobic stability of silage and preventing aerobic spoilage [14]. Generally, organic acid additives to silage should inhibit the growth of yeast and mold without affecting lactic acid fermentation. In experiment 2, the addition of CaP significantly inhibited ethanol and acetic acid compared to the control, while lactate content was not affected. This indicated that CaP is an ideal chemical additive for TMR modulation, which plays an important role in inhibiting fungal growth and avoiding the excessive ethanol fermentation of TMR.

## 5. Conclusions

To effectively utilize apple pomace resources as animal feed, the effects of moisture regulation, microbial inoculants, and chemical additives on the fermentation characteristics and ethanol production of apple pomace-containing TMR were investigated. The homofermentative LAB improved the fermentation quality of TMR more than that of heterofermentative LAB, but neither inhibited ethanol production. Adjusting the moisture of TMR to 50% and adding CaP could produce high-quality TMR and effectively inhibit the excessive production of ethanol. This is important for the effective utilization of apple pomace resources and the promotion of livestock production.

**Author Contributions:** Conceptualization, J.F. and Y.C.; methodology, writing—original draft preparation, J.F.; software, visualization, data curation, Y.C. and Z.D.; project administration, writing—review and editing, funding acquisition, J.F. and Y.C.; methodology, suggestions on revision, Z.D. and Y.C. All authors have read and agreed to the published version of the manuscript.

**Funding:** This research received no external funding.

**Institutional Review Board Statement:** Not applicable.

**Informed Consent Statement:** Not applicable.

**Data Availability Statement:** Data can be shared.

**Conflicts of Interest:** The authors declare no conflict of interest.

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
