# Peer review of "Fermentation Regulation and Ethanol Production of Total Mixed Ration Containing Apple Pomace"

_fermentation, doi:10.3390/fermentation9070692_

Round 1

Reviewer 1 Report

See attached file

Author Response

Thank you very much for evaluating our paper and giving very valuable comment. We have revised the manuscript “Fermentation regulation and ethanol production of total mixed ration containing apple pomace” (fermentation-2444490) carefully based on the comments of Editor Office and Reviewers. In this revised version, changes to our manuscript were all highlighted within the document by using red colored text. We hope that our paper much better quality than before.

The corresponding responses to the Editor and Reviewer are as followings:

Reviewer 1:

1. Introduction: The introduction is the ‘gateway’ to the paper. It was porously written. The authors should create time and write it better using appropriate and latest publications. Lines 46 & 47. Figure 1. Production proportion and estimated world discharge of apple pomace should not appear under introduction, it should be moved to materials and methods section.

Response: Based on your comments, we have revised the introduction section, and put Figure 1 into the "material and method" section.

2. Statistical Analysis: Line 143. This sentence ‘Statistical analyses were performed using SPSS ver. 25 (SAS Institute, Cary, NC, USA)’ should come immediately after the subheading, Statistical analysis.

Response: We have amended this sentence in response to your comments.

3. Results: The results of this research have been casually and loosely reported. This section should be carefully rewritten.

Response: We have reorganised the table and re-described the results section.

4. Discussion: Lines 205-206 This sentence is hard to follow, consider rephrasing it. Suggestion: In addition, apple pomace is rich in sugars such as glucose and fructose, which can promote silage fermentation well.

Response: Yes, we have revised this sentence based on your comment.

5. Line 223 The word ‘information’ appears unnecessary in this sentence, it should be deleted.

Response: Yes, we have deleted the word ‘information’ in this sentence based on your comment.

6. Line 260-263 Knowledgeable audience might find this sentence hard to comprehend, it should be split into two. Suggestion: Previous studies showed that LAB activity decreased in TMR containing green tea residue. In contrast, acetate and ammonia-producing bacteria activity increased with increasing moisture, resulting in decreased lactic acid and increased acetic acid and ammonia-N with increasing moisture [23].

Response: Thanks for your comment, we have splited this sentence into two.

7. Line 304 Researcher does not agree in number with other words in this phrase. It should be written in plural.

Response: Yes, we have written the word “researcher” in the plural based on your comment.

8. Line 332 Ruminant does not agree in number with other words in this phrase. It should be written in plural.

Response: Yes, we have revised the conclusion section and delete this word.

Best regards.

Dr. Yimin Cai

Japan International Research Center for Agricultural Sciences (JIRCAS)

Tsukuba, Ibaraki 305-8686, Japan

Reviewer 2 Report

This manuscript is not a well written paper for a scientific journal and; there are a lot of grammar and English spelling mastics, using nonacademic words, many paragraphs on this manuscript are poorly constructed and need to be rewritten again to improve the clarity.

Line 11: please avoid to use non-academic worlds such as “we”

Figure 1: please delete it, it is scientific journal not a public journal. I think it is not necessary because the introduction section should be summarized and given just the important information related to the topic with the objectives of the study.

Line 71: again please avoid to use “we”

Tables 1 and 2: the tables should be combined together in one table

Table 1: please provide the energy contain in diets

Figure 2, please delete it

Lines 94-110: the experimental design is not clear, please give more details and rewrite again to improve the clarity.

Line 99: what is meant by “The no added TMR was used as control treatment”?

The modes of statistical analyses used are need more explanations.

Lines 145-156: please delete these 2 paragraphs, it is not acceptable to compared between the feedstuffs.

The discussion section is poorly constructed and needs to be rewritten to improve the clarity.

The conclusions are too tentative and needed to rewrite again in term to improve it for summarizing the findings of the main results.

This manuscript is not a well written paper for a scientific journal and; there are a lot of grammar and English spelling mastics, using nonacademic words, many paragraphs on this manuscript are poorly constructed and need to be rewritten again to improve the clarity.

Line 11: please avoid to use non-academic worlds such as “we”

Figure 1: please delete it, it is scientific journal not a public journal. I think it is not necessary because the introduction section should be summarized and given just the important information related to the topic with the objectives of the study.

Line 71: again please avoid to use “we”

Tables 1 and 2: the tables should be combined together in one table

Table 1: please provide the energy contain in diets

Figure 2, please delete it

Lines 94-110: the experimental design is not clear, please give more details and rewrite again to improve the clarity.

Line 99: what is meant by “The no added TMR was used as control treatment”?

The modes of statistical analyses used are need more explanations.

Lines 145-156: please delete these 2 paragraphs, it is not acceptable to compared between the feedstuffs.

The discussion section is poorly constructed and needs to be rewritten to improve the clarity.

The conclusions are too tentative and needed to rewrite again in term to improve it for summarizing the findings of the main results.

Author Response

Thank you very much for evaluating our paper and giving very valuable comment. We have revised the manuscript “Fermentation regulation and ethanol production of total mixed ration containing apple pomace” (fermentation-2444490) carefully based on the comments of Editor Office and Reviewers. In this revised version, changes to our manuscript were all highlighted within the document by using red colored text. We hope that our paper much better quality than before.

The corresponding responses to the Editor and Reviewer are as followings:

Reviewer 2:

1. This manuscript is not a well written paper for a scientific journal and; there are a lot of grammar and English spelling mastics, using nonacademic words, many paragraphs on this manuscript are poorly constructed and need to be rewritten again to improve the clarity.

Response: Thank you for your review, and based on your suggestions, we will be carefully revised grammar and vocabulary in an effort to improve the quality of English. Finally, our manuscript will be re-edited in English.

2. Line 11: please avoid to use non-academic worlds such as “we”.

Response: In the light of your comments, we have amended it by removing the word "we".

3. Figure 1: please delete it, it is scientific journal not a public journal. I think it is not necessary because the introduction section should be summarized and given just the important information related to the topic with the objectives of the study.

Response: Based on your comments, we have removed the figure 1 from the "Introduction" section. In addition, based on the comments of another reviewer, we placed this figure in the "Material and Method" section. Thank you for your understanding.

4. Line 71: again please avoid to use “we”

Response: Yes, in response to your comment, this sentence has been amended and the word "we" has been removed.

5. Tables 1 and 2: the tables should be combined together in one table

Response: Our research team has carefully discussed your suggestion and agreed that Table 1 and 2 represent different experimental elements and are not suitable to be combined into one table. We hope to have your understanding.

6. Table 1: please provide the energy contain in diets

Response: We provied the energy content in Table 1.

7. Figure 2, please delete it

Response: In response to your comments, we have removed Figure 2.

8. Lines 94-110: the experimental design is not clear, please give more details and rewrite again to improve the clarity.

Response: We have rewritten the experimental treatments section to describe the experimental treatments in more detail and hopefully improve the clarity.

9. Line 99: what is meant by “The no added TMR was used as control treatment”? The modes of statistical analyses used are need more explanations.

Response: We have rewritten the test treatment section to describe the control in more detail.

10. Lines 145-156: please delete these 2 paragraphs, it is not acceptable to compared between the feedstuffs.

Response: We have amended this sentence in response to your comments.

11. The discussion section is poorly constructed and needs to be rewritten to improve the clarity.

Response: Yes, we have rewritten the discussion section and try to improve the clarity.

12. The conclusions are too tentative and needed to rewrite again in term to improve it for summarizing the findings of the main results.

Response: Thank you for your comments, we have rewritten the conclusion section to highlight the results of this study.

Best regards.

Dr. Yimin Cai

Japan International Research Center for Agricultural Sciences (JIRCAS)

Tsukuba, Ibaraki 305-8686, Japan

Reviewer 3 Report

Fermentation-2444490 Peer Review Report

Introduction

The introduction is the ‘gateway’ to the paper. It was porously written. The authors should create time and write it better using appropriate and latest publications.

Lines 46 & 47. Figure 1. Production proportion and estimated world discharge of apple pomace should not appear under the introduction; it should be moved to the materials and methods section.

2.3. Statistical Analysis

Line 143. This sentence ‘Statistical analyses were performed using SPSS ver. 25 (SAS Institute, Cary, NC, USA)’ should come immediately after the subheading, Statistical analysis.

3. Results

The results of this research have been casually and loosely reported. This section should be carefully rewritten.

4. Discussion

Lines 205-206 This sentence is hard to follow; consider rephrasing it. Suggestion: In addition, apple pomace is rich in sugars such as glucose and fructose, which can promote silage fermentation well.

Line 223 The word ‘information’ appears unnecessary in this sentence; it should be deleted.

Line 260-263 Knowledgeable audience might find this sentence hard to comprehend; it should be split into two. Suggestion: Previous studies showed that LAB activity decreased in TMR containing green tea residue. In contrast, acetate and ammonia-producing bacteria activity increased with increasing moisture, resulting in decreased lactic acid and increased acetic acid and ammonia-N with increasing moisture [23].

Line 304 Researcher does not agree in number with other words in this phrase. It should be written in the plural.

Line 332 Ruminant does not agree in number with other words in this phrase. It should be written in the plural.

Author Response

Thank you very much for evaluating our paper and giving very valuable comment. We have revised the manuscript “Fermentation regulation and ethanol production of total mixed ration containing apple pomace” (fermentation-2444490) carefully based on the comments of Editor Office and Reviewers. In this revised version, changes to our manuscript were all highlighted within the document by using red colored text. We hope that our paper much better quality than before.

The corresponding responses to the Editor and Reviewer are as followings:

Reviewer 3:

1. The paper aimed to explore the possibility to prepare high-quality TMR using the apple pomace, by moisture adjustment or microbial inoculant and chemical additive, is very interesting.

Response: Yes, we are preparing high-quality apple pomace feed by using microbial inoculants and chemical additives.

2. The authors well depicted the state of the art of the argument which is very important also for the environmental point of view. They used adequate methodologies and well described and discussed their results. Anyway, few corrections and/or additions need to improve the paper quality, as follows:

Response: Thank you for your review and we will make further corrections or additions based on your comments.

3. line 176: please, add also the significant decrease of pH observed using LP inoculant;

Response: We have added the significant decrease of pH observed using LP inoculant in response to your comments.

4. line 217: in my opinion is too much emphatised the fat content of apple pomace. It is 3.4% DM and I do not agree to te sentence “In addition, apple pomace is rich in EE, which can be used as a fat supply source for TMR”.

Response: Yes, we have reworded this sentence.

5. line 226: please add the following sentence: anyway, according to Tudisco et al. (20201), for all the thesis the fermentation was adequate as suggested by the N-NH3 values lower than 7 g/kg total N.

Response: Yes, we have added this sentence based on your comment.

6. References please add the following: Tudisco, R. et al. Effects of Sorghum Silage in Lactating Buffalo Cow Diet: Biochemical Profile, Milk Yield, and Quality. Agriculture 2021, 11, 57.

Response: Yes, we have added this reference in the “references” section based on your comment.

Best regards.

Dr. Yimin Cai

Japan International Research Center for Agricultural Sciences (JIRCAS)

Tsukuba, Ibaraki 305-8686, Japan

Round 2

Reviewer 3 Report

Fermentation-2444490 Peer Review Report, 2nd Round

Introduction

Line 34, the preposition use (‘of’) may be wrong here. Suggestion: Use ‘in’ instead of ‘of’.

Line 38 ‘citru’ is misspelt, check and correct it.

Lines 45, 48, 57 Some spellings (fiber, odor, flavor) are non-British variants. There are others; check the whole text and replace them with British English spellings, for consistency.

Line 51 ‘high moisture’ is missing a hyphen; consider adding it.

Line 62 ‘be used to’ is unnecessary in this sentence and should be removed.

Materials and Methods

Line 74 the first letter ‘d’ in the word ‘december’ should be written in upper case.

Line 79 the word ‘ingredient’ should be in plural.

Line 83 letter ‘t’ in the word ‘the’ should be in upper case.

Line 91 “kilo grams’ should be written as one word.

Line 97 ‘distill’ should be reported in the past tense.

Results

Lines 134-35 This sentence appears difficult to follow; it needs recasting. Suggestion: TMR materials' gross energy (GE) ranged from 4.6 to 5.0 kcal/kg of DM.

Discussion

Line 210 ‘Anyway’ appears informal and unnecessary; consider removing it.

Lines 237-239 This sentence appears hard to understand; it should be rephrased. Suggestion: However, alcoholic fermentation in TMR will cause dry matter and energy loss, and high ethanol content in the feed will negatively impact milk quality and breeding cattle.

These have already been entered above

Author Response

Dear reviewers and editors

Thank you very much for reviewing our paper and providing valuable comments. Based on the comments from the editorial board and reviewers, we have carefully revised our manuscript “Fermentation regulation and ethanol production of total mixed ration containing apple pomace” (fermentation-24444490). In this revised version, our changes to the manuscript are marked in red font. We hope that the quality of our paper is better than before.

The responses to the comments of reviewer 3 are as follows:

1. Introduction: Line 34, the preposition use (‘of’) may be wrong here. Suggestion: Use ‘in’ instead of ‘of’.

Resonse: Yes, we have modified this sentence as per your suggestion.

2. Line 38 ‘citru’ is misspelt, check and correct it.

Resonse: Yes, we have revised.

3. Lines 45, 48, 57 Some spellings (fiber, odor, flavor) are non-British variants. There are others; check the whole text and replace them with British English spellings, for consistency.

Resonse: Yes, we have revised in the whole text.

4. Line 51 ‘high moisture’ is missing a hyphen; consider adding it.

Resonse: Yes, we have added it.

5. Line 62 ‘be used to’ is unnecessary in this sentence and should be removed.

Resonse: Yes, we have deleted.

6. Materials and Methods: Line 74 the first letter ‘d’ in the word ‘december’ should be written in upper case.

Resonse: Yes, we have revised.

7. Line 79 the word ‘ingredient’ should be in plural.

Resonse: Yes, we have changed “ingredient” to “ingredients”.

8. Line 83 letter ‘t’ in the word ‘the’ should be in upper case.

Resonse: Yes, we have revised.

9. Line 91 “kilo grams’ should be written as one word.

Resonse: Yes, we have revised.

10. Line 97 ‘distill’ should be reported in the past tense.

Resonse: Yes, we have revised.

11. Results: Lines 134-35 This sentence appears difficult to follow; it needs recasting. Suggestion: TMR materials' gross energy (GE) ranged from 4.6 to 5.0 kcal/kg of DM.

Resonse: Yes, we have revised.

12. Discussion: Line 210 ‘Anyway’ appears informal and unnecessary; consider removing it.

Resonse: Yes, we have revised.

13. Lines 237-239 This sentence appears hard to understand; it should be rephrased. Suggestion: However, alcoholic fermentation in TMR will cause dry matter and energy loss, and high ethanol content in the feed will negatively impact milk quality and breeding cattle.

Resonse: Yes, we have revised.

The response to the editor's comments is as follows:

We have carefully reviewed the manuscript according to the editor's comments, and we would like to respond to the reviewer's comments as follows:

1. We have carefully revised the manuscript according to the reviewers' comments.

2. We used a linked manuscript for checking and revising.

3. We have carefully checked the references of this thesis, which are all closely related to the research content of this thesis.

4. We edited the original English manuscript in English after revision.

Best regards.

Dr. Yimin Cai

Japan International Research Center for Agricultural Sciences (JIRCAS)

Tsukuba, Ibaraki 305-8686, Japan